# Learning Privileged Degradation Priors for All-in-One Image Restoration

## Abstract

The central challenge in all-in-one image restoration lies in learning degradation-specific priors to effectively modulate a restoration network. Prevailing approaches tackle this by learning representations that can distinguish between degradation types, often via proxy tasks like classification or contrastive learning. However, a representation optimized for discrimination is not necessarily optimal for restoration, leading to a fundamental objective mismatch. To address this, we introduce the Learning Using Privileged Information (LUPI) paradigm. Our method employs a teacher network granted privileged access to both degraded and clean images during training, allowing it to learn a prior directly guided by the final restoration quality. This process yields an ideal, inherently "restoration-aware" prior, which a student network—observing only the degraded input—is then trained to approximate. The learned prior dynamically modulates a restoration backbone for adaptive recovery, enabling our unified model to achieve state-of-the-art performance on benchmarks. Visualizations confirm the learned prior space is semantically structured, revealing intrinsic relationships between degradation types and effectively distinguishing their intensities. The code will be made publicly available upon acceptance of the paper.

## 1 Introduction

Image restoration, the process of recovering a high-quality clean image from a degraded observation, is a fundamental problem in computer vision. In recent years, deep learning has achieved remarkable success in task-specific restoration, with specialized models excelling at individual tasks such as denoising (Zhang et al., 2017), deraining (Chen et al., 2023), or deblurring (Lai et al., 2016). However, real-world degradations are often diverse and complex, making the approach of training a separate model for each specific corruption type computationally expensive and impractical for real-world deployment. This limitation has spurred significant research into all-in-one image restoration, which seeks to address a wide spectrum of degradations with a single, unified model.

A naive approach to training such a unified model—simply mixing data from all tasks—often leads to performance degradation due to task conflict, where the optimization for one task (e.g., sharpening for deblurring) can interfere with another (e.g., smoothing for denoising) (Potlapalli et al., 2023; Duan et al., 2024). To mitigate this, the dominant paradigm in recent literature has been to learn a degradation-aware prior that can dynamically modulate a shared restoration backbone. Early and influential approaches in this direction focused on learning representations at a category-level. For instance, methods based on contrastive learning (Li et al., 2022), explicit classification (Hu et al., 2025), or text instructions (Conde & Geigle, 2024) all aim to map a given corrupted image to a discrete degradation type (e.g., "denoising" vs. "deraining"). While effective at separating distinct categories, this strategy is inherently inflexible as it struggles to capture the continuous variation of degradation intensity within the same class (e.g., light versus heavy noise), often resulting in a one-size-fits-all guidance that is sub-optimal for precise restoration.

Recognizing this limitation, more recent works have shifted towards learning instance-level, adaptive priors. Methods like AdaIR (Cui et al., 2025) and MoCE-IR (Zamfir et al., 2025) implicitly learn a representation from the degraded image by optimizing the final restoration loss in an end-to-end fashion. While this instance-level adaptivity is a significant step forward, we argue these methods face a new fundamental challenge: the ambiguity of the supervision signal. Because the

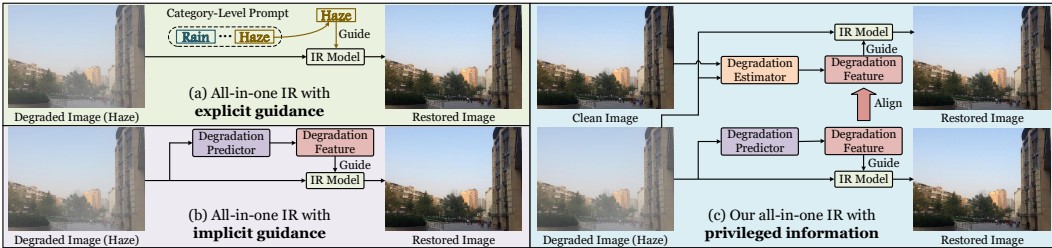

Figure 1: A conceptual comparison of different paradigms for learning degradation priors in All-in-One image restoration. (a) Explicit Guidance: Methods that rely on predefined, discrete prompts (e.g., category labels like "Haze") to guide the restoration network. (b) Implicit Guidance: Methods that learn a degradation representation end-to-end from only the degraded image. (c) Our Method (Privileged Learning): We introduce a framework where a teacher Estimator, granted privileged access to both clean and degraded images, learns a high-quality degradation prior. A student Predictor is then trained to approximate this prior using only the degraded image, providing a restoration-aware guidance signal at inference time.

prior-generation module only ever sees the corrupted input, it must attempt to solve an extremely difficult inverse problem: to disentangle the unknown degradation from the unknown clean content using only a single, mixed signal. The guidance it receives—a scalar restoration loss propagated back through a deep network—is often insufficient to resolve this ambiguity. Essentially, the network is trapped in a "chicken-and-egg" dilemma: it needs a good prior to restore the image well, but it can only learn a good prior if the restoration network is already effective enough to provide a clear gradient.

To resolve this fundamental ambiguity and break the "hicken-and-egg" cycle, we reframe the problem from a different theoretical standpoint: Learning Using Privileged Information (LUPI) Vapnik & Vashist (2009). Instead of attempting to learn a prior from an incomplete and mixed signal, we propose a paradigm that learns this prior under the guidance of an oracle. The core idea is to provide the model with extra, "privileged" information during the training phase that is unavailable at test time. In our context, the ground-truth clean image $I_c$ serves as this powerful privileged information. By having access to both the degraded input $I_d$ and the clean target $I_c$, a "teacher" network is uniquely positioned to directly infer the true nature of the degradation transformation, thus learning a prior that is inherently "restoration-aware".

Our framework materializes this paradigm through a teacher-student architecture. During training, a teacher estimator learns a degradation prior by observing both $I_d$ and its clean counterpart $I_c$. Crucially, this teacher's learning is supervised directly by the final restoration loss, ensuring the resulting prior is optimized for the restoration task, not a proxy. Subsequently, a student predictor, which only ever sees the degraded input, is trained to approximate this ideal, privileged prior via a distribution alignment loss. This critical step transfers the knowledge from the teacher space to the student, enabling the student to generate a high-quality, restoration-aware prior for any unseen degraded image at inference time. The learned prior is then used to dynamically modulate a restoration backbone for adaptive recovery.

- We introduce the Learning Using Privileged Information (LUPI) paradigm to the all-in-one image restoration domain as a direct solution to this problem.
- We propose a novel framework that effectively learns "restoration-aware" degradation priors by leveraging clean images as privileged information and using the final restoration quality as the direct supervision signal.
- Our proposed LUPI model achieves state-of-the-art performance across multiple benchmarks, demonstrating the practical superiority of our proposed approach.

## 2 RELATED WORK

**Single-Task Image Restoration**. Image restoration aims to restore a clean image from its degraded observation. Existing approaches are commonly categorized into prior-based and data-driven meth-

ods. Prior-based methods reduce the solution space via physical or statistical assumptions—e.g., (He et al., 2009) employs the dark channel prior for dehazing. But it typically generalizes poorly to complex real-world degradations. In contrast, data-driven methods learn mappings from degraded observations to their clean counterparts using large-scale training data and exhibit superior generalization compared with prior-based approaches. These methods employ convolutional neural networks (CNNs) and have achieved strong task-specific results in denoising (Zhang et al., 2017), deraining (Chen et al., 2023), dehazing (Ren et al., 2016), and low-light enhancement Guo et al. (2017). More recently, to model long-range dependencies and global context, Transformer architectures have been introduced for image restoration; representative methods (Song et al., 2023; Tsai et al., 2022) report substantial gains on dehazing and deblurring. Despite these advances, most of the methods are designed for a specific type of degradation and exhibit limited cross-degradation generalization. More recent models, such as Restormer (Zamir et al., 2022), Uformer (Wang et al., 2022), and SFHformer (Jiang et al., 2024), show competitive performance across multiple restoration tasks, yet they typically require task-specific training and maintaining separate checkpoints per degradation. In real-world scenarios, however, images often suffer from various or compounded degradations, rendering single-task models less practical for deployment due to their restricted generalization and increased maintenance overhead.

**All-in-One Image Restoration**. All-in-one image restoration aims to handle multiple degradation types (e.g., denoising, deraining, dehazing, and deblurring) within a unified framework, facilitating practical deployment in real-world scenarios with diverse degradations. Early designs adopt task-specific encoders and decoders to cope with multiple degradations (Chen et al., 2021), but they typically assume the degradation type is known, which limits practicality. Recent studies have proposed various strategies to learn degradation-aware representations, enabling adaptation to varying restoration tasks without requiring prior knowledge of the degradation type. A pioneering work, Air-Net (Li et al., 2022), employs contrastive learning to extract degradation representations and guide image restoration without explicit degradation labels. AdaIR (Cui et al., 2025) mines frequency-domain cues and performs feature modulation to adaptively handle different degradations. Inspired by prompt learning, several methods introduce learnable prompts that encode degradation context and modulate the restoration backbone accordingly (Potlapalli et al., 2023; Luo et al., 2024; Yang et al., 2024; Duan et al., 2024). For example, PromptIR (Potlapalli et al., 2023) integrates a visual prompt block that implicitly infers the degradation condition and dynamically guides restoration across diverse types, albeit with non-trivial parameter overhead. With the emergence of vision–language models (VLMs), InstructIR (Conde & Geigle, 2024) further explores human-written instructions to direct restoration. While instruction-driven paradigms improve adaptivity, they often rely on large-scale pre-trained language models, inflating system complexity and computational cost. In contrast, we introduce the LUPI paradigm, in which a privileged teacher learns a restoration-aware prior, and a student predicts this prior from the degraded input at test time—requiring neither degradation labels nor language models.

## 3 METHODOLOGY

### 3.1 OVERALL FRAMEWORK

The central thesis of our work is that a powerful all-in-one image restoration model requires a degradation prior explicitly optimized for the restoration task itself. Existing paradigms often falter due to an *objective mismatch* or *supervision ambiguity*. To overcome these limitations, we introduce a novel framework grounded in the Learning Using Privileged Information (LUPI) paradigm. Our framework is built around two key components: a **Degradation Prior Estimator**, which generates a latent vector representing the degradation, and a **Modulated Restoration Network**, which uses this prior to adaptively restore the corrupted image. The core of our contribution lies in how this prior is learned. We propose a two-stage, teacher-student training strategy that leverages the ground-truth clean image as privileged information, as depicted in Figure 2. First, in **Stage 1**, we train a privileged **teacher system** where an expert estimator ($D_T$) sees both the clean ($I_c$) and degraded ($I_d$) images to produce an ideal, "restoration-aware" prior $\mathbf{d}_T$. This prior is learned by directly optimizing the final output of a teacher restoration network ($R_T$). Subsequently, in **Stage 2**, we train a practical **student system**. Here, a student predictor ($D_S$) learns to generate a similar high-quality prior $\mathbf{d}_S$ by observing only the degraded image, which is achieved by forcing its output to align with that of

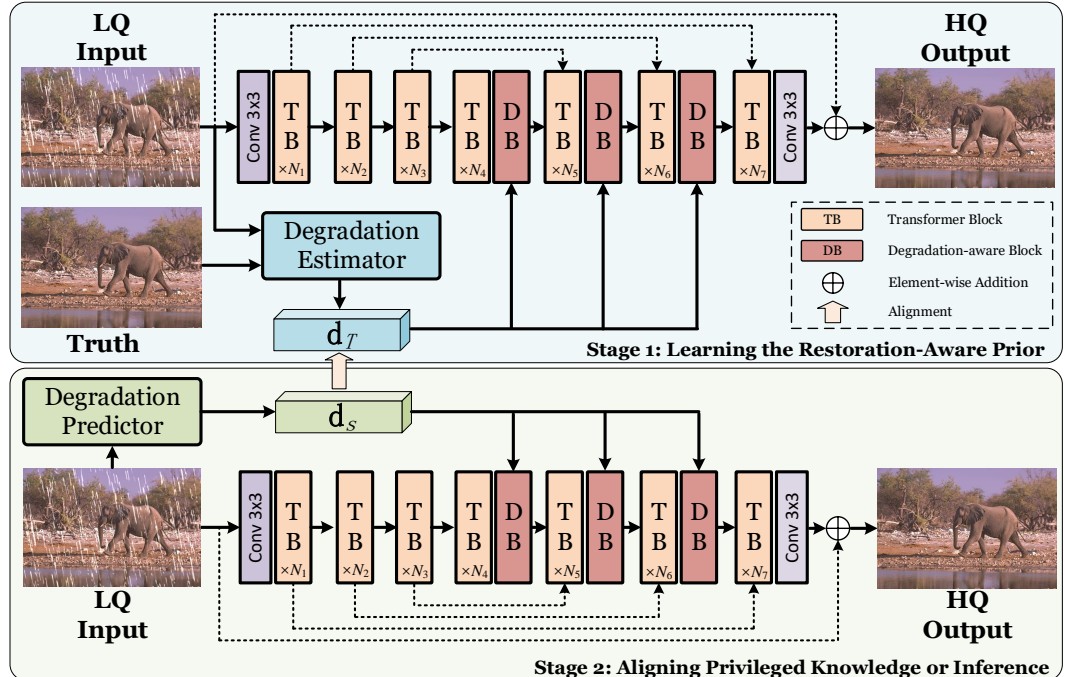

Figure 2: **An overview of our proposed framework based on the Learning Using Privileged Information (LUPI) paradigm.** Our method consists of two training stages. **In Stage 1 (top)**, a privileged estimator ($D_T$) utilizes both the degraded ($I_d$) and clean ($I_c$, referred to as Truth) images to generate an ideal, restoration-aware prior ($\mathbf{d}_T$). This prior is learned by optimizing the end-to-end restoration quality of the teacher network. **In Stage 2 (bottom)**, a degradation predictor ($D_S$) is trained to generate a degradation prior ($\mathbf{d}_S$) from only the degraded input, by aligning it with the frozen privileged prior $\mathbf{d}_T$. At **inference time**, the process is identical to Stage 2's forward pass: the predicted prior $\mathbf{d}_S$ dynamically modulates the restoration network's Degradation-aware Blocks (DBs) to produce the final restored output.

the frozen teacher estimator. This strategy allows us to first define what an optimal prior is under ideal conditions, and then teach a practical model to produce it.

## 3.2 ARCHITECTURAL COMPONENTS

Our architectural designs are detailed in Figure 3. We build upon established modules to emphasize that our performance gains stem from the training paradigm.

**Modulated Restoration Network ($R$).** The overall architecture of our restoration network ($R_T$ and $R_S$) follows the successful paradigm of recent state-of-the-art methods such as PromptIR (Potlapalli et al., 2023) and AdaIR (Cui et al., 2025), which consists of a powerful restoration backbone and a feature modulation mechanism. As illustrated in Figure 2, our backbone is a U-Net whose core component is the Transformer Block (TB) from Restormer (Zamir et al., 2022); the detailed structure of the TB can be found in the appendix A.1. The feature modulation is achieved by our proposed **Degradation-aware Block (DB)**, which injects the degradation prior generated by our **Degradation Estimator** ($D_T$) or **Predictor** ($D_S$). Following the effective design choices of PromptIR and AdaIR, we strategically place these DBs in the deeper stages of the U-Net decoder. This allows the prior to modulate features at a higher semantic level for more effective guidance. We will now detail the three core components of our degradation-aware design: the Privileged Degradation Estimator $D_T$, the Degradation Predictor $D_S$, and the Degradation-aware Block (DB).

**Privileged Degradation Estimator ($D_T$).** The degradation estimator $D_T$ in Figure 3 (a) extracts the privileged prior $\mathbf{d}_T$ from the ($I_d, I_c$) pair. It employs a Siamese-like architecture where both

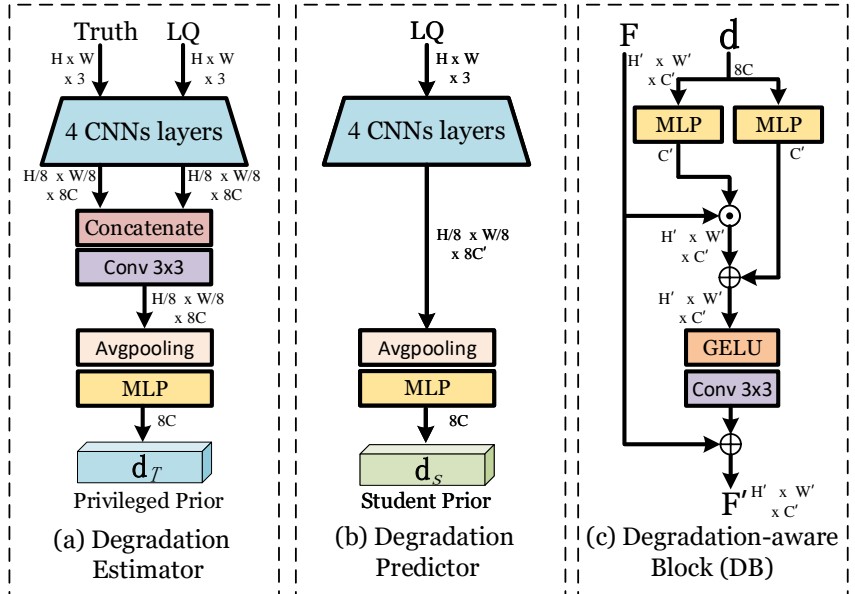

Figure 3: **Detailed architectures of our key network components. (a) The Privileged Estimator** ($D_T$) employs a Siamese-like design with a shared-weight CNN encoder. It takes both the degraded (LQ) and clean (Truth) images as input, concatenates their features, and passes them through a fusion module and an MLP head to produce the privileged prior $\mathbf{d}_T$. **(b) The Student Predictor** ($D_S$) mirrors a single branch of the teacher's architecture, learning to predict a similar prior $\mathbf{d}_S$ from only the degraded input. **(c) The Degradation-aware Block (DB)** is the mechanism for injecting the learned prior. The prior vector is transformed by an MLP into a channel-wise scaling vector, which then element-wise multiplies the feature map that has been processed by a convolutional block.

inputs pass through four shared-weight CNN layers, mapping them to feature spaces of size $H/8 \times W/8$. These features are then concatenated, passed through a $3 \times 3$ convolutional fusion layer, and finally projected by an average pooling layer and an MLP into the final prior vector $\mathbf{d}_T \in \mathbb{R}^{8C}$.

**Degradation Predictor** ($D_S$). The student predictor $D_S$ in Figure 3 (b) mirrors a single branch of the teacher. It takes only the degraded image $I_d$ as input, passes it through the same four CNN layers to extract features, and then uses an average pooling layer and an MLP to predict the student prior $\mathbf{d}_S \in \mathbb{R}^{8C}$. This architectural consistency simplifies the knowledge alignment in Stage 2.

**Degradation-aware Block (DB).** The **Degradation-aware Block (DB)** is responsible for injecting the learned degradation-aware prior into the network to modulate its features. For the sake of efficiency, we adopt a simple and lightweight approach based on the Feature-wise Linear Modulation (FiLM) mechanism (Perez et al., 2018). As illustrated in Figure 3c, the block's operation is direct and straightforward. Given the output feature map $F \in \mathbb{R}^{H' \times W' \times C'}$ from a corresponding decoder stage and the degradation prior $\mathbf{d} \in \mathbb{R}^{8C}$, the prior is first transformed by a lightweight MLP into a scaling vector $\gamma \in \mathbb{R}^{C'}$ and a shifting vector $\beta \in \mathbb{R}^{C'}$:

$$[\gamma, \beta] = \mathrm{MLP}(\mathbf{d}). \tag{1}$$

These parameters then directly modulate the input feature map $F$ before it is passed through the block's main path. A final residual connection (He et al., 2016) ensures information flow. The entire operation can be summarized as:

$$F' = F + \mathrm{Conv}_{3 \times 3}(\mathrm{GELU}(\gamma \odot F + \beta)), \tag{2}$$

where $F' \in \mathbb{R}^{H' \times W' \times C'}$ is the output of DB.

### 3.3 LEARNING WITH PRIVILEGED INFORMATION: A TWO-STAGE APPROACH

With all components defined, we now detail our two-stage training procedure.

**Stage 1: Learning the Restoration-Aware Prior.** In this stage, the teacher estimator $D_T$ and restoration network $R_T$ are trained jointly. The system is optimized using an L1 loss between the restored image $I_r$ and the ground truth $I_c$:

$$\mathcal{L}_{\text{teacher}} = \|R_T(I_d, D_T(I_d, I_c)) - I_c\|_1. \tag{3}$$

The gradients from this loss shape the prior $\mathbf{d}_T$ to be explicitly optimized for the restoration task.

**Stage 2: Distilling Privileged Knowledge.** In this stage, the weights of $D_T$ are frozen. We then jointly train the student predictor $D_S$ and a new student restorer $R_S$. The training objective is a composite loss:

$$\mathcal{L}_{\text{student}} = \mathcal{L}_{\text{recon}} + \lambda_{\text{align}}\mathcal{L}_{\text{align}}, \tag{4}$$

where $\lambda_{\text{align}}$ is a balancing hyperparameter. The reconstruction loss $\mathcal{L}_{\text{recon}}$ is the L1 loss on the final student output. The alignment loss $\mathcal{L}_{\text{align}}$ encourages $\mathbf{d}_S$ to match $\mathbf{d}_T$ via a Kullback-Leibler (KL) divergence loss.

### 3.4 INFERENCE

At inference time, the teacher system is discarded. The final model consists only of the student predictor $D_S$ and restorer $R_S$. Given a degraded input $I_d$, the model computes $\mathbf{d}_S = D_S(I_d)$ and then produces the restored image $I_o = R_S(I_d, \mathbf{d}_S)$.

## 4 EXPERIMENTS

To validate the efficacy of our proposed LUPI, we conduct extensive experiments on the challenging task of all-in-one image restoration. We first detail the experimental setup, including the standard benchmarks and our implementation specifics. We then present a comprehensive comparison of our method against state-of-the-art competitors under both 3-task and 5-task settings. For all quantitative evaluations, we employ the Peak Signal-to-Noise Ratio (PSNR) and the Structural Similarity Index (SSIM) as our primary metrics, where higher values indicate better restoration quality.

### 4.1 EXPERIMENTAL SETUP

**Datasets.** To ensure a fair and direct comparison with recent state-of-the-art methods (Potlapalli et al., 2023; Cui et al., 2025), we evaluate our model on two widely adopted multi-task benchmarks. The first is a **3-task benchmark** comprising **Denoising**, **Deraining**, and **Dehazing**. For denoising, we synthesize training data by adding Gaussian noise ($\sigma \in \{15, 25, 50\}$) to the BSD400 (Arbelaez et al., 2010) and WED (Ma et al., 2016) datasets, and evaluate on the BSD68 benchmark (Martin et al., 2001). For deraining and dehazing, we use the Rain100L (Wenhan Yang & Yan, 2017) and SOTS (Li et al., 2018) datasets, respectively. The second, more challenging **5-task benchmark** extends this setup with two additional tasks: **Deblurring** on the GoPro dataset (Nah et al., 2017) and **Low-Light Enhancement** on the LOL-v1 dataset (Wei et al., 2018).

**Implementation Details.** We implement our framework in PyTorch and conduct all experiments on two NVIDIA L40 GPUs. Consistent with the architecture described in Section 3, our restoration network is a U-Net with four encoder levels and three decoder levels, and the number of transformer blocks across the seven stages is set to [4, 6, 6, 8, 6, 6, 8]. The dimension of the learned degradation prior is set to 512 (i.e. $8C$ in Figure 3 ). We use the AdamW optimizer (Loshchilov & Hutter, 2017) for all training. During training, we extract patches of size $128 \times 128$ and apply random horizontal flipping and rotation for data augmentation.

**Training Strategy.** Our framework is trained end-to-end following the proposed two-stage paradigm. **In Stage 1**, we train the complete teacher system ($D_T$ and $R_T$) for 150 epochs, using the L1 reconstruction loss as the sole optimization objective, as defined in Equation 3. **In Stage 2**, after freezing the teacher estimator's weights, we train the student system ($D_S$ and $R_S$) for another 150 epochs. For this stage, we use the composite loss function defined in Equation 4, which combines the L1 reconstruction loss with the KL divergence-based alignment loss to transfer the teacher's knowledge. For both training stages, the initial learning rate is set to $2 \times 10^{-4}$ and is gradually decayed to zero using a cosine annealing schedule.

Table 1: Quantitative comparison (PSNR / SSIM) for all-in-one restoration on three tasks. The best results are in **bold**, and the second-best are underlined.

| Method | Dehazing | Deraining | Denoising on BSD68 | | | Average |
|---|---|---|---|---|---|---|
| | SOTS | Rain100L | $\sigma = 15$ | $\sigma = 25$ | $\sigma = 50$ | |
| AirNet (Li et al., 2022) | 27.94 / 0.962 | 34.90 / 0.967 | 33.92 / 0.933 | 31.26 / 0.888 | 28.00 / 0.797 | 31.20 / 0.910 |
| PromptIR Potlapalli et al. (2023) | 30.58 / 0.974 | 36.37 / 0.972 | 33.98 / 0.933 | 31.31 / 0.888 | 28.06 / 0.799 | 32.06 / 0.913 |
| Art-PromptIR (Wu et al., 2024) | 30.83 / 0.979 | 37.94 / 0.982 | 34.06 / 0.934 | 31.42 / 0.891 | 28.14 / 0.801 | 32.49 / 0.917 |
| InstructIR (Conde & Geigle, 2024) | 30.22 / 0.959 | 37.98 / 0.978 | 34.15 / 0.933 | 31.52 / 0.890 | 28.30 / 0.804 | 32.43 / 0.913 |
| PromptIR-TUR (Wu et al., 2025) | 31.17 / 0.978 | 38.57 / 0.984 | 34.06 / 0.932 | 31.40 / 0.887 | 28.13 / 0.797 | 32.67 / 0.916 |
| AdaIR (Cui et al., 2025) | 31.06 / 0.980 | 38.64 / 0.983 | 34.12 / 0.935 | 31.46 / 0.892 | 28.19 / 0.802 | 32.69 / 0.918 |
| VLU-Net (Zeng et al., 2025) | 30.71 / 0.980 | **38.93** / 0.984 | 34.13 / 0.935 | 31.48 / 0.892 | 28.23 / 0.804 | 32.70 / 0.919 |
| MoCE-IR (Zamfir et al., 2025) | 31.34 / 0.979 | 38.57 / 0.984 | 34.11 / 0.932 | 31.45 / 0.888 | 28.18 / 0.800 | 32.73 / 0.917 |
| **Ours (LUPI)** | **31.86 / 0.983** | 38.92 / **0.985** | **34.23 / 0.937** | **31.58 / 0.894** | **28.33 / 0.807** | **32.98 / 0.921** |

Table 2: Quantitative comparison (PSNR / SSIM) for all-in-one restoration on five tasks. Best results are in **bold**, second-best are underlined. Note that for denoising, we report results for $\sigma = 25$ following standard practice in this setting.

| Method | Dehazing | Deraining | Denoising | Deblurring | Low-Light | Average |
|---|---|---|---|---|---|---|
| | SOTS | Rain100L | $BSD68_{\sigma=25}$ | GoPro | LOL | |
| AirNet (Li et al., 2022) | 21.04 / 0.884 | 32.98 / 0.951 | 30.91 / 0.882 | 24.35 / 0.781 | 18.18 / 0.735 | 25.49 / 0.847 |
| PromptIR (Potlapalli et al., 2023) | 25.20 / 0.931 | 35.94 / 0.964 | 31.17 / 0.882 | 27.32 / 0.842 | 20.94 / 0.799 | 28.11 / 0.883 |
| Gridformer (Wang et al., 2024) | 26.79 / 0.951 | 36.61 / 0.971 | 31.45 / 0.885 | 29.22 / 0.884 | 22.59 / 0.831 | 29.33 / 0.904 |
| InstructIR Conde & Geigle (2024) | 27.10 / 0.956 | 36.84 / 0.973 | 31.40 / 0.873 | 29.40 / 0.886 | 23.00 / 0.836 | 29.55 / 0.908 |
| Transweather-TUR Wu et al. (2025) | 29.68 / 0.966 | 33.09 / 0.952 | 30.40 / 0.869 | 26.63 / 0.815 | 23.02 / 0.838 | 28.56 / 0.888 |
| AdaIR (Cui et al., 2025) | 30.53 / 0.978 | 38.02 / 0.981 | 31.35 / 0.889 | 28.12 / 0.858 | 23.00 / 0.845 | 30.20 / 0.910 |
| VLU-Net (Zeng et al., 2025) | 30.84 / 0.980 | 38.54 / 0.982 | 31.43 / 0.891 | 27.46 / 0.840 | 22.29 / 0.833 | 30.11 / 0.905 |
| MoCE-IR (Zamfir et al., 2025) | 30.48 / 0.974 | 38.04 / 0.982 | 31.34 / 0.887 | **30.05 / 0.899** | 23.00 / 0.852 | 30.58 / 0.919 |
| **Ours (LUPI)** | **31.00 / 0.981** | **39.20 / 0.986** | **31.55 / 0.894** | 29.46 / 0.886 | **23.66 / 0.865** | **30.97 / 0.922** |

## 4.2 QUANTITATIVE AND QUALITATIVE COMPARISONS

To comprehensively evaluate our framework, we benchmark our LUPI-based model against state-of-the-art (SOTA) methods on both a 3-task and a more challenging 5-task all-in-one restoration benchmark. The quantitative results, presented in Table 1 and Table 2, demonstrate the clear superiority of our approach, which achieves the best overall performance in both settings. On the 5-task benchmark, for instance, our method surpasses the strong MoCE-IR baseline by a significant **0.39 dB** in average PSNR. These quantitative improvements are visually substantiated by our qualitative results in Figure 4. The visual comparisons reveal our model's enhanced ability to restore fine-grained textures while faithfully removing degradations. For example, in the deraining result, our method recovers the subtle skin textures of the subject more effectively than competing methods, while in the dehazing example, it restores the vibrant colors of the street signs with higher fidelity. Collectively, these strong quantitative and qualitative results validate the effectiveness of learning a restoration-aware prior through our proposed LUPI paradigm. More qualitative comparisons can be found in the appendix A.3, and a comparison of model runtime efficiency can be found in the appendix A.2

## 4.3 ABLATION STUDY

To thoroughly validate the effectiveness of our proposed framework and analyze the contribution of its key components, we conduct a series of ablation studies on the 3-task benchmark. We investigate three primary aspects: the impact of our LUPI-based training paradigm, the architectural design of the privileged teacher estimator, and the characteristics of the learned prior space.

**Impact of the LUPI Framework.** This core ablation evaluates the fundamental contribution of our privileged learning strategy. We compare our full model against two degraded variants. The first, termed **w/o Privileged information**, removes the Stage 1 training entirely. In this setting, the student system is trained end-to-end from scratch using only the L1 reconstruction loss, representing a standard instance-adaptive model. The second variant, **w/o Degradation Predictor**, is further simplified by removing the adaptive module altogether, degenerating into a single restoration network.

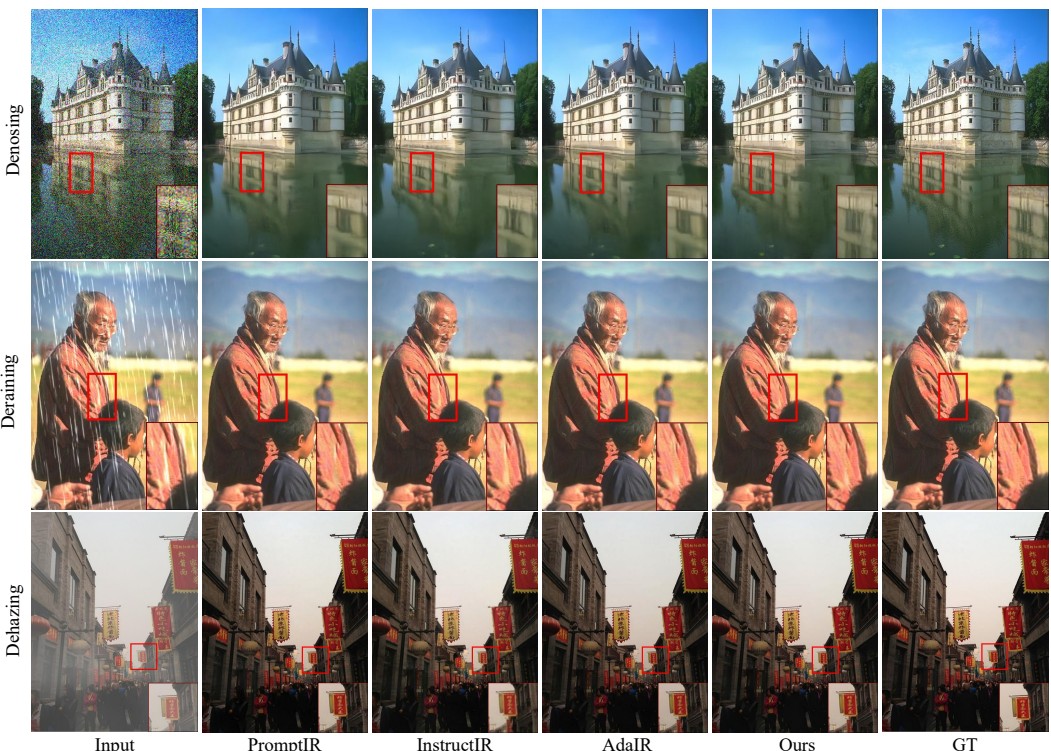

Figure 4: Qualitative comparison on the 3-task benchmark (denoising, deraining, and dehazing). Zoom in for the best view.

The results in Table 3 show that our full LUPI significantly outperforms both variants. The substantial performance drop in the **w/o Privileged Teacher** setting validates our central hypothesis that the guidance from a privileged, restoration-aware teacher is crucial for overcoming the limitations of learning from ambiguous signals. Furthermore, the poor performance of the **w/o Degradation Predictor** variant confirms the necessity of a dynamic mechanism for handling diverse degradations.

Table 3: Impact of the LUPI framework. Average performance is reported.

| Estimator ($D_T$) | Predictor ($D_S$) | PSNR ↑ | SSIM ↑ |
|---|---|---|---|
| X | X | 31.94 | 0.907 |
| X | ✓ | 32.07 | 0.909 |
| ✓ | ✓ | **32.98** | **0.921** |

Table 4: Design of the estimator ($D_T$). Average performance is reported.

| Method | PSNR ↑ | SSIM ↑ |
|---|---|---|
| Input Addition | 32.75 | 0.918 |
| Input Concate | 32.80 | 0.920 |
| **Siamese (Ours)** | **32.98** | **0.921** |

**Design of the Privileged Teacher Estimator.** We investigate how different strategies for processing the privileged information ($I_d$ and $I_c$) in the degradation estimator affect final performance. We compare our proposed **Siamese** design against two simpler alternatives: **Input Concatenation**, where the images are concatenated along the channel dimension before being fed to the estimator, and **Input Addition**, where the two images are element-wise added. As shown in Table 4, our Siamese architecture yields the best performance. This design allows the network to extract comparable features before fusion, which is more effective for identifying degradation characteristics. Concatenating the inputs performs reasonably well but is slightly inferior, while simple addition leads to a more performance drop, validating our choice of the Siamese architecture.

**Visualization of the Prior Space.** To intuitively understand the properties of the restoration-aware prior, we analyze the latent space of the degradation predictor ($D_S$) in Figure 5. The analysis

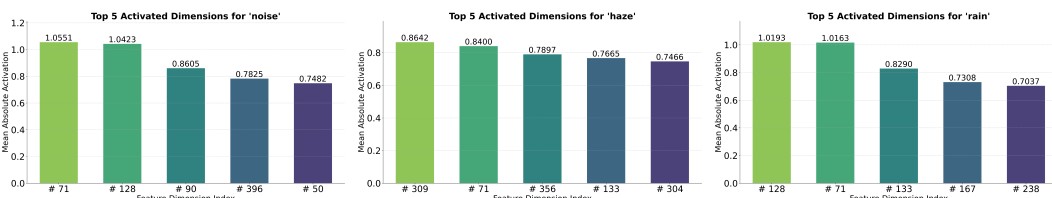

(a) Top 5 activated prior dimensions for different degradation types (noise, haze, rain).

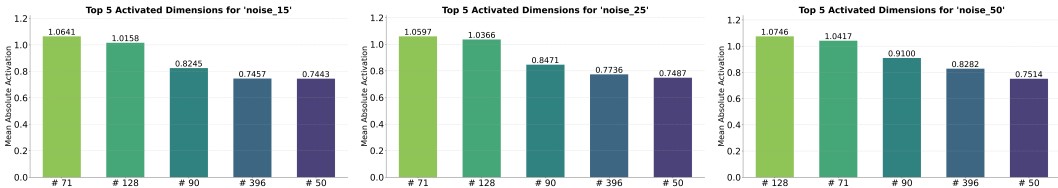

(b) Top 5 activated prior dimensions for different noise intensities ($\sigma = 15, 25, 50$).

Figure 5: **Visualization of the learned degradation prior space.** **(a)** For different degradation types, the model learns physically-grounded representations. Note the significant overlap between "noise" and "rain", which share properties with additive corruptions. The commonly activated dimension #71 suggests an encoding for general attributes like *local occlusion*, while "haze" activates a distinct pattern. **(b)** For different noise intensities, the prior demonstrates a disentangled encoding of degradation *type* and *intensity*. The set of activated dimensions remains stable (identifying the degradation as 'noise'), while their relative magnitudes shift to encode the severity, enabling precise, intensity-aware restoration.

reveals that our framework learns a highly structured and interpretable prior space that captures the underlying physical nature of degradations, rather than just their surface-level appearance.

First, the model learns to group tasks based on their physical similarities. As shown in Figure 5 (a), the priors for deraining and denoising exhibit a strong overlap in their most activated dimensions (e.g., #71 and #128), reflecting their shared properties with additive corruptions. Conversely, dehazing, a spatially-varying degradation, activates a distinct set of dimensions. This demonstrates that the model learns physically-grounded representations.

This structured representation extends to a finer granularity. A deeper analysis within the denoising task (Figure 5b) reveals that the model has learned a partially disentangled representation of degradation *type* and *intensity*. We observe that a remarkably stable set of dimensions is used to represent noise regardless of its severity, forming a canonical representation for the 'noise' category. Critically, within this stable set, the model encodes the continuous intensity by modulating the relative activation magnitudes. For instance, the activation of dimension #128 increases with the noise level (from 1.016 at $\sigma = 15$ to 1.075 at $\sigma = 50$). The emergence of such an interpretable prior is a direct benefit of our LUPI framework, whose unambiguous supervision signal guides the model to learn the true underlying factors of degradation, leading to its robust and precise restoration capabilities.

## 5 CONCLUSION

In this work, we addressed the challenge of learning effective degradation priors for all-in-one image restoration by introducing the Learning Using Privileged Information (LUPI) paradigm to resolve the objective mismatch and supervision ambiguity in existing methods. Our LUPI framework allows a privileged teacher to learn an optimal, "restoration-aware" prior from both clean and degraded images, which a student network then learns to predict. Our method achieves state-of-the-art performance on extensive multi-task benchmarks. Furthermore, we demonstrate that the learned prior space is not a black box but is highly structured and interpretable, capable of capturing the physical similarities between degradation types and even disentangling their categorical type from their continuous intensity.

REPRODUCIBILITY STATEMENT

To ensure the reproducibility of our results, we have included our source code in the an anonymous repository https://anonymous.4open.science/r/lupi-C4BC/.

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

## A APPENDIX

### A.1 TRANSFORMER BLOCK ARCHITECTURE

The Transformer Block (TB) used in our restoration network is adopted directly from the design of Restormer (Zamir et al., 2022), as illustrated in Figure 6. Each block is composed of two primary sub-modules: a Multi-Dconv Head Transposed Attention (MDTA) module for global feature aggregation, followed by a Feed-Forward Network (FFN) for feature transformation. The key innovation of this block lies in the MDTA, which computes attention across feature channels rather than spatial locations, making it an efficient and effective component for high-resolution image restoration tasks.

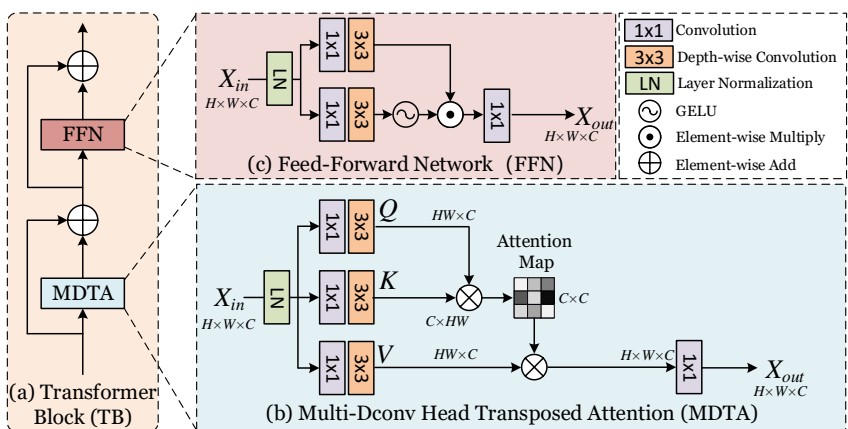

Figure 6: **Architecture of the Transformer Block (TB).** The block consists of two main components in sequence: (a) The Multi-Dconv Head Transposed Attention (MDTA) module. (b) a Feed-Forward Network (FFN).

### A.2 EFFICIENCY AND PERFORMANCE ANALYSIS

To provide a comprehensive view of our model's practical utility, we benchmark its performance and efficiency against several key methods. We specifically choose the Restormer baseline (Zamir et al., 2022), PromptIR (Potlapalli et al., 2023), and AdaIR (Cui et al., 2025) for this comparison, as their architectures are most similar to ours: they all combine a powerful Restormer-based backbone

Table 5: Comparison of model performance and efficiency on the 3-task benchmark (average scores). Our method is compared against the Restormer baseline and other state-of-the-art methods. The best performance is highlighted in **bold**.

| Method | Params | GFLOPS | Memory | Latency | Throughput | PSNR | SSIM |
|---|---|---|---|---|---|---|---|
| Restormer (Zamir et al., 2022) | 26.13M | 154.88G | 676.00MB | 50.67ms | 19.73FPS | 31.94 | 0.907 |
| PromptIR (Potlapalli et al., 2023) | 35.59M | 172.71G | 720.35MB | 55.03ms | 18.17FPS | 32.06 | 0.913 |
| AdaIR (Cui et al., 2025) | 28.78M | 161.76G | 686.15MB | 62.74ms | 15.94FPS | 32.69 | 0.918 |
| **Ours (LUPI)** | 31.93M | 163.20G | 698.69MB | 52.55ms | 19.03FPS | **32.98** | **0.921** |

with a dedicated module for degradation-aware feature modulation. All metrics were evaluated on a single NVIDIA RTX 4090 GPU, using an input resolution of $3 \times 256 \times 256$. The reported latency is the average of 100 inference runs following a sufficient warmup period to ensure stable results. The detailed comparison is presented in Table 5.

The results highlight that our LUPI-based framework achieves a superior balance between performance and efficiency. While our model (31.93M params) is moderately larger than the Restormer baseline (26.13M params), this increased complexity is a direct result of incorporating the degradation-aware modules, which proves to be a worthwhile trade-off, yielding a significant performance gain of over 1.0 dB in PSNR.

More importantly, when compared to other state-of-the-art adaptive methods, our model demonstrates compelling efficiency. It is notably more lightweight and faster than PromptIR across all metrics. The comparison with AdaIR is particularly insightful. Despite having approximately 10% more parameters, our model's latency of 52.55ms is about 16% lower than AdaIR's 62.74ms. This suggests that our Degradation-aware Block (DB) has a more hardware-friendly architectural design that translates to better practical inference speed. In summary, our LUPI framework delivers state-of-the-art restoration accuracy without compromising, and in some cases even improving upon, the practical deployability of comparable methods.

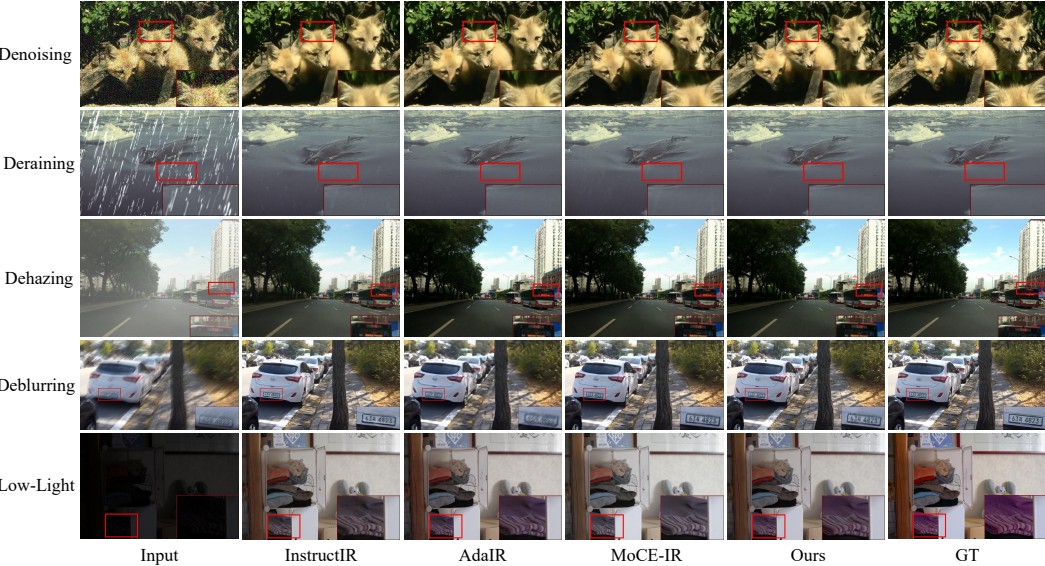

Figure 7: Qualitative comparison on the 5-task benchmark: denoising, deraining, dehazing, deblurring, and low-light enhancement. Zoom in for best view.

## A.3 QUALITATIVE RESULTS ON THE 5-TASK BENCHMARK

To further validate the generalization capability of our model, we provide qualitative comparisons on the more challenging 5-task benchmark in Figure 7. This benchmark tests the model's ability

to handle a wider and more diverse set of degradations, including denoising, deraining, dehazing, deblurring, and low-light enhancement.

As the visual results show, our method demonstrates consistently superior performance across all five tasks. For low-light enhancement, our model effectively brightens the scene while accurately restoring colors and suppressing noise in dark regions, avoiding the color casts or artifacts present in other methods. In the deblurring example, our approach successfully recovers sharp details, particularly on the license plate of the vehicle, with high fidelity. For denoising and deraining, our model excels at removing the respective corruptions while better preserving fine-grained textures, such as the fur on the cat and the surface of the runway. Finally, in the dehazing task, our result is visually more pleasing, with more natural contrast and color balance.

These strong qualitative results across a diverse set of tasks further substantiate the effectiveness of our LUPI framework. The learned restoration-aware prior is versatile enough to guide the restoration network through a wide variety of complex degradations, leading to consistently high-quality outputs.

## A.4 THE USE OF LARGE LANGUAGE MODELS (LLMS)

During the preparation of this paper, we employed a Large Language Model (LLM) to assist with improving the language and readability of the text. The primary use of the LLM was for proofreading, including correcting grammatical errors and refining sentence structure to enhance clarity. We confirm that the LLM was not used for research ideation, developing the methodology, conducting experiments, analyzing results, or drawing conclusions. All intellectual contributions and scientific claims are solely those of the authors.

