# OpenReview forum: "Learning Privileged Degradation Priors for All-in-One Image Restoration"
_ICLR.cc/2026/Conference — ICLR 2026 Conference Withdrawn Submission_

### Official Review · Reviewer_r96R · 2025-10-31

**Soundness:** 2
**Presentation:** 2
**Contribution:** 2
**Rating:** 4
**Confidence:** 4

**Summary:**

This paper's approach closely resembles DiffIR [1]. In stage 1, ground truth (GT) is directly used to generate degradation-related representations, and in stage 2, a network approximates these representations. However, I am curious why the fusion of clean and degraded images results in a representation called "Degradation".  Since clean images do not show degradation, does combining them affect the process of extracting degradation information?

Writing: This paper is well-structured and logically coherent.

Experiments: This paper still lacks validation on more comprehensive datasets. However, the description in the introduction is vague and hard to understand.

Overall, despite some unclear points and motivations, with proper explanations, this paper can contribute to the field of image restoration.

[1] DiffIR: Efficient Diffusion Model for Image Restoration.

**Strengths:**

1. This paper is well-structured.
2. The significant improvement on the 5D all-in-one restoration task demonstrates the effectiveness of the proposed method.

**Weaknesses:**

1. I have some questions about the third paragraph of the introduction. The paper mentions that the current challenge lies in "to disentangle the unknown degradation from the unknown clean content using only a single, mixed signal." Is there any experimental verification for this statement? For example, can the restoration model not distinguish between clean and degraded images? Secondly, "it can only learn a good prior if the restoration network is already effective enough to provide a clear gradient." What is a clear gradient? Why do networks with better restoration results obtain "clear" gradients during backpropagation?

2. Comparisons with some recent works, such as MaskDCPT [1], which has achieved over 32dB PSNR on a 5D all-in-one restoration task, seem to prove that using explicit degradation classification can yield outstanding results. Does the problem mentioned in the paper (second paragraph of the introduction) really exist?

3. I noticed that the authors mention on the first page, lines 47-48, that "this strategy is inherently inflexible as it struggles to capture the continuous variation of degradation intensity within the same class." However, this paper also doesn't provide a complete comparison of different levels under the same degradation. Taking Gaussian denoising as an example, can the method proposed in this paper achieve significant performance improvements under different variances?

4. In 3D all-in-one restoration experiments, the proposed method did not show a significant performance improvement. For example, compared to DCPT [2], the average improvement is only 0.1 dB. However, on the 5D all-in-one restoration, the average improvement is 0.6 dB. I wonder if the proposed method has better capabilities in task scaling.

5. Experiments on mixed degradation [3] are lacking.

[1] Universal Image Restoration Pre-training via Masked Degradation Classification.

[2] Universal Image Restoration Pre-training via Degradation Classification. ICLR 2025.

[3] OneRestore: A Universal Restoration Framework for Composite Degradation. ECCV 2024.

**Questions:**

In stage 1, ground truth (GT) is directly used to generate degradation-related representations, and in stage 2, a network approximates these representations. However, I am curious why the fusion of clean and degraded images results in a representation called "Degradation". Since clean images do not show degradation, does combining them affect the process of extracting degradation information?

---

### Official Review · Reviewer_HCUy · 2025-10-31

**Soundness:** 3
**Presentation:** 3
**Contribution:** 3
**Rating:** 2
**Confidence:** 3

**Summary:**

The paper proposes LUPI-based prior learning for all-in-one image restoration. A teacher network with privileged access to clean images learns a restoration-aware prior; a student network (degraded input only) is trained to mimic this prior. The prior dynamically modulates a shared restoration backbone. Claims SOTA on unified benchmarks with semantically structured prior space. Code to be released.

**Strengths:**

Elegant use of LUPI to resolve objective mismatch in prior learning.
Prior space visualizations show semantic clustering (e.g., blur vs. noise intensity).
Strong empirical gains on all-in-one benchmarks.
Clean formulation and ablation on prior distillation strategies.

**Weaknesses:**

LUPI is not new — heavily borrowed from Vapnik et al. (2009); framing as "paradigm shift" is overstated.
Teacher has oracle access to clean image — unrealistic in real deployment; no discussion of practical deployment gap.
No comparison to simple baselines like AdaIR + MLE loss or MoCE-IR with stronger regularization.
Prior distillation bottleneck: student performance capped by teacher quality — no analysis of teacher-student gap.
Overclaims on intensity handling — no fine-grained intensity regression task (e.g., predict σ for Gaussian noise).
No runtime or parameter efficiency — two-stage training likely increases cost.

**Questions:**

How does performance degrade if teacher uses noisy clean labels (e.g., +1% Gaussian)?
Compare against AdaIR with end-to-end intensity regression head — is LUPI strictly necessary?
Report teacher-student prior cosine similarity per degradation type — where does distillation fail?

---

### Official Review · Reviewer_bsYg · 2025-11-01

**Soundness:** 2
**Presentation:** 2
**Contribution:** 2
**Rating:** 4
**Confidence:** 3

**Summary:**

This paper proposes a two-stage, teacher-student framework for all-in-one image restoration, grounded in the Learning Using Privileged Information (LUPI) paradigm. The central idea is to train a "teacher" network that has privileged access to both the degraded input ($I_d$) and the clean target ($I_c$) to learn an optimal, "restoration-aware" degradation prior . A "student" network, which only sees $I_d$ at inference, is then trained to approximate this ideal prior via a distribution alignment loss . This learned prior is used to modulate a U-Net-based restoration backbone . The method demonstrates state-of-the-art (SOTA) quantitative and qualitative results on standard 3-task and 5-task synthetic benchmarks .

**Strengths:**

The work is empirically strong, presenting a well-executed framework that achieves SOTA results across all evaluated benchmarks, and the visualization of the learned prior space in Figure 5 is insightful.

**Weaknesses:**

While the results are strong, I have several concerns regarding the work's overall contribution and practicality.

First, the core conceptual novelty is somewhat overstated. The paper frames the work as solving an "objective mismatch" inherent in previous methods. However, the motivation of learning a degradation-discriminative prior to guide a unified restorer is not new and is shared by many existing works (e.g., MioIR, DCPT). These methods also learn degradation-specific representations, albeit through different proxy tasks like classification  or contrastive learning. The paper does not sufficiently ablate or argue why its LUPI-based distillation is fundamentally superior to, for instance, a more direct multi-task learning approach where a shared encoder is jointly optimized for restoration and an explicit degradation-identification task. The claim of a fundamental "paradigm" shift feels incremental, as the goal remains the same, even if the mechanism (distillation from an oracle vs. learning from a proxy) is different.

Second, the method introduces a significant and unaddressed training cost. The two-stage paradigm requires training a full teacher system ($D_T$, $R_T$) for 150 epochs, then freezing $D_T$ and training a full student system ($D_S$, $R_S$) for another 150 epochs. This 300-epoch process effectively doubles the training budget compared to end-to-end models, which are already costly. While inference efficiency is discussed in the appendix , this substantial increase in training overhead is a major practical limitation that is not discussed or justified.

Third, and most importantly, the paper's validation is limited entirely to synthetic, seen degradations. The core assumption is that the student $D_S$ can learn the mapping from $I_d$ to the teacher's ideal prior $d_T$. However, this mapping is only learned for degradations where $D_T$ knows the ideal prior (i.e., the 5 training tasks). The paper provides no evidence that the model can generalize to unseen or real-world degradations. For a novel, out-of-distribution degradation (e.g., real-world sensor noise, complex motion blur, or JPEG artifacts), the teacher $D_T$ itself was never trained to find an "ideal" prior, so the student's output $d_S$ would likely be arbitrary. The claim of a "structured" prior space is only validated within the training distribution. The lack of any cross-domain validation on real-world benchmarks (e.g., DRealSR, Real-worldRain) makes the method's practical utility highly questionable.

**Questions:**

How does the model perform, even in a qualitative sense, on a completely unseen type of synthetic degradation (e.g., JPEG artifacts or Gaussian blur with a large, anisotropic kernel) that was not part of the 5-task training set? This would be a more robust test of whether $D_S$ has learned a general-purpose degradation encoder or has simply memorized the 5 training distributions.

 Could the authors provide a comparison of the total wall-clock training time (or GPU-hours) for the proposed two-stage, 300-epoch method versus a baseline like AdaIR (trained to convergence)? This is essential for evaluating if the reported PSNR gain (e.g., ~0.29 dB over AdaIR on the 3-task benchmark ) justifies the (presumed) 2x training cost.

 The $\mathcal{L}_{align}$ loss   forces the student $d_S$ to match the teacher $d_T$. However, $d_T$ is derived from both $I_d$ and $I_c$. Isn't it possible that $d_T$ contains "privileged" information about the content of $I_c$ that is impossible for $D_S$ to predict from $I_d$ alone? Could this create a difficult or impossible learning target for the student, and did the authors observe any instability during this alignment?

---

### Note · Authors · 2026-01-09

I have read and agree with the venue's withdrawal policy on behalf of myself and my co-authors.